# 3-Ferrocenyl-estra-1,3,5 (10)-triene-17-one: Synthesis, Crystal Structure, Hirshfeld Surface Analysis, DFT Studies, and Its Binding to Human Serum Albumin Studied through Fluorescence Quenching and In Silico Docking Studies

**DOI:** 10.3390/molecules28166147

**Published:** 2023-08-20

**Authors:** Mariola M. Flores-Rivera, José A. Carmona-Negrón, Arnold L. Rheingold, Enrique Meléndez

**Affiliations:** 1Department of Chemistry, University of Puerto Rico, P.O. Box 9019, Mayaguez, PR 00681, USA; mariola.flores@upr.edu (M.M.F.-R.); jose.carmona@upr.edu (J.A.C.-N.); 2Department of Chemistry and Biochemistry, University of California–San Diego, Urey Hall 5128, 9500 Gilman Drive, La Jolla, CA 92093, USA; arnold.rheingold@ucsd.edu

**Keywords:** ferrocene, crystal structure, Hirshfeld surface, DFT, fluorescence quenching

## Abstract

3-ferrocenyl-estra-1,3,5 (10)-triene-17-one (**2**), [Fe(C_5_H_5_)(C_24_H_25_O_3_)], crystallizes in the monoclinic space group C2. The cyclopentadienyl (Cp) rings adopt a nearly eclipsed conformation, and the Cp plane is tilted by 87.66° with respect to the substituted phenyl plane. An average Fe-C(Cp) bond length of 2.040(13) Å was determined, similar to the one reported for ferrocene. Hirshfeld surfaces and two-dimensional fingerprint plots were generated to analyze weak intermolecular C-H···π and C-H···O interactions. Density functional theory studies revealed a 1.15 kcal/mol rotational barrier for the C3-O1 single bound. Fluorescence quenching studies and in silico docking studies suggest that human serum albumin forms a complex with **2** via a static mechanism dominated by van der Waals interactions and hydrogen bonding interactions.

## 1. Introduction

Conventional metal-based drugs, such as cisplatin and derivatives, have proven successful in treating several types of cancers such as colon, lung, and ovarian cancer [1,2]. The cytotoxic activity of these derivatives is achieved through binding to the purine bases on nuclear DNA. However, these therapeutic agents lack selectivity between healthy and cancerous tissue, causing side effects detrimental to the body [3].

Köpf-Maier and his co-workers first reported the anticancer properties of ferrocenium in 1984 [4]. This organometallic compound forms radical oxygen species (ROS) that cause oxidative damage to DNA, inducing cell apoptosis [5]. Our research group has successfully incorporated ferrocene at estrogen’s rings A and D, showing antiproliferative effects in vitro on hormone-dependent MCF-7 and hormone-independent MDA-MB-231 breast cancer cell lines [6,7,8,9,10,11].

In 2011, the synthesis, electrochemistry, and biological activity of 3-ferrocenyl-estra-1,3,5 (10)-triene-17-one (**2**) on breast cancer (MCF-7) and colon cancer (HT-29) cell lines were reported [10]. However, the crystal structure of this ferrocene–hormone complex was not elucidated. To circumvent this problem, we used a different synthetic strategy, which provided a clean product able to crystallize as a single crystal. In this article, we present the synthesis and compare the crystal structure of **2** (CCDC: 2075475) with 3-hydroxy-1,3,5(10)-estratrien-17-one (**1**) (CCDC: 228769) [12], 3-ferrocenyl-estra-1,3,5 (10)-triene-17β-ol (**3**) (CCDC:1842145) [8], and 3-benzyl-estra-1,3,5 (10)-triene-17β-ol (4) (CCDC:127103) [13]. The structures of these complexes are shown in Figure 1. C-H···π and C-H···O interactions were analyzed by generating Hirshfeld surfaces and two-dimensional fingerprint plots. Density functional theory studies were employed to gain structural insights into the rotational barrier for the C3-O1 single bond. Human serum albumin interactions with the title compound were studied through fluorescence quenching and in silico docking studies.

## 2. Results and Discussion

### 2.1. Synthesis and Characterization

The syntheses of 3-ferrocenyl-estra-1,3,5(10)-triene-17-one (**2**) and 3-ferrocenyl-estra-1,3,5 (10)-triene-17β-ol (**3**) were first reported by our group in 2011 [10]. These ferrocene–hormone complexes were obtained using ferrocenecarbonylchloride as the starting material. In 2019, a new synthetic route to compound **3** was reported [8], which doubled the yield percentage and allowed the formation of crystals of the synthesized compound. Motivated by these results, we revisited the synthesis of **2** to develop a cleaner and more efficient synthetic method. 

Ferrocene–hormone complex **2** was synthesized using estrone and fluorocarbonyl ferrocene as the starting materials. The reaction scheme is shown in Figure 2. The reaction was performed using 4-(dimethylamino)pyridine in dry CH_2_Cl_2_ under reflux for 12 h. Single crystals were obtained using the liquid–vapor diffusion technique. This compound was characterized using Nuclear Magnetic Resonance Spectroscopy (NMR; Appendix A), Fourier Transform Infrared Spectroscopy (FT-IR; Appendix A), and X-ray Diffraction (XRD). 

The IR spectroscopy results showed two ν(C=O) absorption bands in the range of 1707–1726 cm^−1^, which corresponded to the carbonyl groups on C17 of estrone and the ester linker group. In the ^1^H NMR spectrum, a singlet (5H integration) was observed at 4.31 ppm, corresponding to the unsubstituted Cp of the ferrocene, and two multiplets in the range of 4.96–4.50 ppm, confirming the substitution of ferrocene. The ^13^C NMR spectrum displayed a signal at 220.9 ppm, corresponding to the carbonyl carbon at C17, and another signal at 170.11 ppm, corresponding to the carbonyl group of the ester. All the spectra matched those previously reported for **2**.

### 2.2. Crystal Structure

The crystal structure of 3-ferrocenyl-estra-1,3,5 (10)-triene-17-one (**2**) is shown in Figure 3. This ferrocene–hormone complex was found to crystallize in a monoclinic unit cell and a C2 space group, similar to the crystal structure of 3-ferrocenyl-estra-1,3,5 (10)-triene-17β-ol (**3**) previously reported by our group [8]. The most relevant bond lengths and angles for the title compound crystal structure are shown in Table 1. Figure 4 shows a conformational comparison of **2**, **3**, and **4**.

The bond length between C3 and O1 of **2** was 1.408(2) Å, similar to **3** and **4**. On the other hand, the distance between C19 and O1 was 1.364 (3)Å, slightly deviating from that reported for **4** (1.337 (3) Å). The C3-O1-C19 angles of **2** and **3** were somewhat contracted compared to that of **4,** 115° vs. 119° (Figure 4A,B), due to the difference in bulkiness between the two substituents (ferrocenyl vs. benzyl groups). These substituents also affected the angle between the cyclopentadienyl plane and the aromatic group, 88.51° for **2** and 87.66° for **3**, whereas for the benzyl group on 3-benzyl-estra-1,3,5 (10)-triene-17β-ol (**4**), the angle was 62.37° (Figure 4B). The two cyclopentadienyl rings adopted a nearly eclipsed conformation in structures **2** and **3**. The average Fe-C(Cp) bond length was 2.040(13) Å for **2** and 2.043(13) Å for **3**. The Fe-C(Cp)_subt_ bond distance was found to be the same in both structures.

The distance between the C17-O2 double bond and the hormone carbonyl of **2** was 1.202 (3) Å, similar to that of estrone **1** (1.219 (2) Å). As reported for other steroids, the angle between C17, C13, and C14 suffered a deviation from the 108° expected for a five-member ring. In **2**, an angle of 100.7 (**2**)° was determined. 

#### 2.2.1. Supramolecular Features

In the crystal structure of 3-ferrocenyl-estra-1,3,5 (10)-triene-17-one (**2**), molecules were linked by C-H···O hydrogen bonds and C-H···π interactions. The bond lengths and angles of these interactions are described in Table 2. The molecular packing of the ferrocene-hormone complex **2** is shown in Figure 5. There was an important C-H···π interaction taking place between the C-H of one Cp ring and the Cp π cloud of the neighboring molecule. There were also two C-H···π interactions taking place between one Cp π cloud and two hydrogens from the hormone moiety (H11B and H14). Additionally, there was hydrogen bonding between the C-H of the Cp and the carbonyl group of the next neighboring molecule. The same ferrocene-binding interphase of **2** was observed in the crystal structure of **3** (Figure 5B). Neither C-H···O nor C-H···π interactions were found in the benzyl substituent of **4**. 

Since the ferrocene substitution took place on O1 of the estrogen moiety, no head-to-tail hydrogen bonding interactions were observed for **2**, as reported for **1** [12].

#### 2.2.2. Hirshfeld Surface Analysis

Hirshfeld surfaces (HS) and 2D fingerprint plots were generated for the title complex using CrystalExplorer21 [14]. The calculated surface had a volume of 568.12 Å^3^. To analyze and visualize the crystal packing, the HS was mapped over the distance external to the surface (*de*), shape index (*S*), and curvedness (*C*). The results of this analysis are shown in Figure 6. The HS mapped over *de* highlights the closest contacts of the neighboring molecules with respect to the generated surface (orange regions). Curvedness divides the surface into bonding or contact regions, and low values (blue) and high values (green) of *C* represent edges and flat regions, respectively. Meanwhile, the shape index (*S*) is a qualitative measurement highly sensitive to small changes in a surface’s shape. The donor and acceptor regions are shown as blue bumps and red hollows, respectively [15,16].

Figure 6A illustrates the inspection process for a C-H···C contact between neighboring molecules. The orange region on the *de*-mapped surfaces denotes a close contact between the H_21_ and the cyclopentadienyl centroid (Cg_2_). Meanwhile, the *C*-mapped surface shows a low-curvature region. Finally, the *S*-mapped property reveals a negative electrostatic potential region on the surface, confirming the presence of a C-H···π interaction [17].

The Hirshfeld surface was also mapped over the normalized function of distances, *de* and *di* (*dnorm*). This property uses blue (*dnorm* > 0), white (*dnorm* = 0), and red (*dnorm* < 0) to identify long, at van der Waals separations, and short contacts, respectively [18]. The closest contacts for the ferrocene–hormone complex surface (Figure 6B) corresponded to C-H···π and C-H···O interactions. Two-dimensional fingerprint plots were generated to provide a visual representation of the frequency of certain contacts within the surface. Figure 6C shows the highlighted contributions of C-H···C and C-H···O interactions. For the title complex surface, the following contributions of interatomic contacts were determined: H···H (63.4%), C-H···O/O···H-C (18.9%) C···H-C/C-H···C (17.3%), C···C (0.2%), and C···O/O···C (0.2%).

### 2.3. Density Functional Theory Study

The crystallographic structures of the 3-ferrocenyl-estra-1,3,5 (10)-triene-17-one (**2**), 3-ferrocenyl-estra-1,3,5 (10)-triene-17β-ol (**3**), and 3-benzyl-estra-1,3,5 (10)-triene-17β-ol (**4**) derivatives showed different dihedral angles Φ_1_ (defined as C2-C3-O1-C19). We performed a DFT relaxed potential energy surface (PES) scan to study the energetic barrier governing the rotation of the C3-O1 single bond and its relationship with the substituent (benzene vs. ferrocene). The crystal structures of **2** and **4** were used as starting points for the gas-phase optimization. 

PES scans were performed using the optimized structures. The dihedral angle Φ_1_ was restricted and rotated by 5° over 72 steps. Figure 7 and Figure 8 show the PES scan results for complexes **2** and **4**, respectively. For ferrocene derivative **2**, the lowest energy conformer corresponds to structure **a** (Figure 7B), with an angle of 358.3° (−1.70°). By adopting this angle, the substituted Cp of ferrocene became coplanar with respect to the aromatic ring of the estrone moiety. Structures **b** (93.30°) and **d** (263.30°) were conformers with maximum energy (Figure 7B). In both structures, the ferrocene-substituted Cp was perpendicular to the aromatic ring. The rotational energy was calculated by subtracting the minimum from the maximum value. For ferrocene–hormone conjugate **2**, it was approximately 1.15 kcal/mol.

The dihedral angle Φ_1_ in the complex **2** crystal structure was 97.5°. When comparing this structure with the local maximum at 93.30°, we can see that the conformations adopted in both structures were very similar. This means that the energy cost of exceeding the 1.15 kcal/mol rotational barrier can be easily offset by the C-H···π and C-H···O interactions found in the crystal packing of **2**. This was also observed in the crystal structure of ferrocene–hormone complex **3**. This suggests that this conformation provides a more stable crystal packing under these crystallizing conditions, maximizing the binding opportunities of these ferrocene–hormone complexes.

A similar behavior was observed in the PES scan of benzyl–hormone conjugate **4** (Figure 8A). Conformers that corresponded to the local maxima (**b** and **d**) were found to have the plane of the benzyl substituent perpendicular to the aromatic ring plane in the hormone moiety. The structure with the highest energy exhibited a dihedral angle of 89.96° (Figure 8B). Meanwhile, the conformers with lower energies were found to have coplanar aromatic planes (**a** and **c**). The lowest-energy conformer (**c**) had a dihedral angle of 183.30°. The rotational barrier of benzyl–hormone complex **4** was 0.803 kcal/mol.

### 2.4. Human Serum Albumin Binding Interaction

Human serum albumin (HSA) is the most prevalent protein in plasma. This protein has two drug-binding sites (sites I and II) and seven fatty acid binding pockets (BPs), which exhibit different characteristics (polarity, BP volume, length, etc.) [19]. Therefore, HSA has the ability to bind to a wide range of compounds with high affinity and plays an important role as the main carrier of various bioactive molecules such as fatty acids, hormones, metals, nucleic acids, and drugs [19,20]. Given the significance of understanding the transport mechanisms of compounds in blood plasma, HSA is a critical protein target for the development of new drugs. In light of this, we decided to study the formation of a complex between HSA and ferrocene–hormone complex **2** through quenching fluorescence, as well as its possible binding pose and affinity through in silico docking studies.

#### 2.4.1. Fluorescence Quenching Studies

HSA is considered to have intrinsic fluorescence, mainly originating from the Trp214 amino acid residue located in subdomain IIA [21]. When HSA binds to a ligand, this fluorescence is quenched in response to this interaction. Quenching mechanisms are classified based on their dependence on temperature as dynamic or static [22]. If the quenching constants increase proportionally with the temperature, the mechanism is classified as dynamic. This temperature increase also increases the diffusion coefficient. Meanwhile, a static quenching mechanism is related to the formation of complexes and binding interactions between the quencher and fluorophore. In this case, a decrease in the quenching constants is observed when the temperature is increased. A third possible scenario involves a combination of dynamic and static mechanisms.

The interaction between HSA and **2** was studied through fluorescence quenching, and the results of this analysis are shown in Figure 9. The fluorescence spectra of HSA were recorded upon excitation at 280 nm (Figure 9A). The emission intensity at 336 nm decreased with the increasing concentration of compound **2**. To elucidate this quenching mechanism, the experimental data were analyzed using the Stern–Volmer equation [23]:$$\frac{F_{0}}{F}=1+k_{q}\tau_{0}\left[Q\right]=1+K_{sv}\left[Q\right]$$
where $F_{0}$ is the fluorescence intensity in the absence of the quencher, F is the fluorescence intensity in the presence of the quencher, $\left[Q\right]$ is the concentration of the quencher, $K_{sv}$ is the Stern–Volmer quenching constant, $k_{q}$ is the biomolecular quenching constant, and $\tau_{0}$ is the lifetime of the fluorescence in the absence of the quencher ($10^{-8}s^{-1}$ for a biomolecule). Stern–Volmer plots were generated for the **2**–HSA interaction at 293, 298, and 303 K (Figure 9B). Linear fitting of the experimental data provided the Stern–Volmer constants listed in Table 3. 

The Stern–Volmer plots showed good linearity at all three temperatures, which suggests that only one mechanism occurred. The Stern–Volmer constant $K_{sv}$ decreased with increasing temperature, indicating a static mechanism. In a static mechanism, $k_{q}$ is larger than the maximum scattering collision quenching constant $k_{d}$ (2.0 × 10^10^ L/mol s). The formation of the complex through a static mechanism was confirmed by the biomolecular quenching constant, which was much greater than 2.0 × 10^10^ L/mol s. 

To understand this binding interaction, binding constant values were determined using fluorescence intensity data. The equilibrium of complex formation between the ligand and protein is given by [24,25]:$$log\frac{F_{0}-F}{F}=logK+nlog\left[Q\right]$$
where $F_{0}$ and F are the fluorescence intensities in the absence and presence of the quencher, respectively; $\left[Q\right]$ is the concentration of the quencher; K is the binding constant; and n is the number of equivalent binding sites to which the ligand binds. Plots were generated for this equation (Figure 9C). All three temperatures exhibited good linearity, and the results are listed in Table 4. The values of n for HSA were approximately equal to one, indicating that **2** binds to one binding site during the interaction. K values measure the ability of a drug to bind to a protein. Binding constants in the range of 1–15 × 10^4^ L/mol correspond to a moderate affinity [26]. In this case, the K values for the **2**–HSA interaction suggest a reversible interaction and a moderate affinity with the protein.

Finally, the thermodynamic parameters governing this interaction were calculated using the van’t Hoff equation [21]:$$\ln K=-\frac{\Delta H}{RT}+\frac{\Delta S}{R}$$
where K is the association constant for the protein–ligand complex at a given temperature, T is the temperature, R is the gas constant (8.314 J/mol K), ΔH is the enthalpy change, and ΔS is the entropy. In this case, K is correlated with the Stern–Volmer ($K_{sv}$) constant. A van’t Hoff plot was generated for the **2**–HSA interaction (Figure 9D); ΔH was calculated from the slope, and ΔS was calculated from the intercept given by the equation of the line from the linear fitting. Using these values, the change in the free energy (ΔG) was calculated for each temperature using the following equation:ΔG=ΔH−TΔS

The results of this analysis are listed in Table 5. Ross and Subramanian developed a model to explain ligand–proteins interaction using thermodynamic parameters and crystallographic data [27]. Following this model, it can be concluded that van der Waals interactions and hydrogen bonding were mainly responsible for the formation of the **2**–HSA complex because ΔS°<0 and ΔH°<0. This bonding interaction was found to be spontaneous because ΔG = −30 kJ/mol. 

#### 2.4.2. Docking Studies

In silico docking studies were performed to understand the **2**–HSA complex formation observed in the fluorescence quenching studies. As previously mentioned, human serum albumin has two drug-binding sites (sites I and II) and seven fatty acid binding pockets [28]. Site I is a large and flexible binding site located in subdomain IIA. Drug-binding site II is located in subdomain IIIA, possesses a less complicated structure, and is characterized by a hydrophilic interior [19]. Site II is known to be frequently occupied by hydrophobic nonsteroidal anti-inflammatory drugs such as ibuprofen, flurbiprofen, and diflunisal. It is also known to show a particular affinity with metal complexes. A previous computational study reported by our group contemplated the binding possibility of ferrocene complexes with all the HSA binding pockets, including sites I and II [29]. Among them, site II also showed the highest affinity with the ferrocene–hormone conjugate.

Figure 10 shows the docking pose results for **2** in HSA binding site II. The **2**–HSA complex showed a scoring function with a binding affinity of -15.2 kcal/mol. A hydrogen bond formed between the hydroxy group in SER489 and the carbonyl group of the ester linker of **2** (Figure 10B). The binding pose of ferrocene–hormone complex **2** suggests that the **2**–HSA complex was mainly formed through hydrophobic interactions with the amino acid residues LEU387, TYR411, VAL415, LEU423, VAL426, LEU430, LEU453, LEU457, LEU460, VAL473, and PHE488. The TYR411 and PHE488 residues appeared to bind to the ferrocene derivative through C-H···π stacking interactions (Figure 10C,D). The site I docking study resulted in a scoring function with a binding affinity of −12.4 kcal/mol (Appendix A).

The molecular picture provided by the in silico docking studies was consistent with the results observed in the fluorescence study. The quenching analysis suggested the formation of a **2**–HSA complex through van der Waals and hydrogen bonding interactions. This ferrocene–hormone complex is highly hydrophobic, and given the characteristics of site II, it is more likely that it could be transported through the blood plasma by binding to this drug-binding site.

## 3. Materials and Methods

### 3.1. Synthesis and Characterization

Estrone (26.01 mg, 0.15 mmol) was added to a solution of fluorocarbonylferrocene [30] (26.01 mg, 0.1 mmol) and N,N-dimethylpyridin-4-amine (18.35 mg, 0.15 mmol) in 1.5 mL dry CH_2_Cl_2_. The reaction mixture was subjected to refluxing for 12 h and then allowed to cool at room temperature. An equal volume of water (1.5 mL) was added, resulting in the organic phase separating, and the aqueous phase was then extracted using EtOAc. The product was dried over CaCl_2_, filtered, and the solvent was evaporated. The crude orange solid was purified using column chromatography with CH_2_Cl_2_:EtOAc (7:3) as the mobile phase. Single crystals suitable for the X-ray diffraction measurements were obtained using the vapor diffusion technique with CHCl_3_:pentane. ^1^H NMR (500 MHz, CDCl_3_) δ (ppm): 7.34 (d, 1H, ^3^J = 8.44 Hz; H_1_), 6.96 (d, 1H, ^3^J = 8.57 Hz; H_2_), 6.92 (s, 1H; H_4_), 4.96 (s, 2H; Cp_1a_), 4.50 (s, 2H; Cp_1b_), 4.31 (s, 5H; Cp_2_), 2.96 (m, 2H; H_6αβ_), 2.53 (dd, ^3^J = 18.92, 1H, ^3^J = 8.69; H_16β_), 2.44 (m, 1H; H_11α_), 2.33 (t, 1H; H_9_), 2.19−1.98 (m, 4H; H_16α_, H_12β_, H_7β_, H_15α_), 1.65−1.49 (m, 6H; H_11β_, H_8_, H_15β_, H_7α_, H_12α_, H_14_), 0.94 (s, 3H, 18; H18abc). ^13^C NMR (125 MHz, CDCl_3_), δ (ppm): 220.9 (C=O), 170.7, 149.0, 138.2, 137.3, 126.6, 121.9, 119.1, 72.1, 70.9, 70.5, 70.2, 50.7, 48.2, 44.4, 38.3, 36.1, 31.8, 29.7, 26.6, 26.0, 21.8, 14.1. IR (KBr, cm−1): 3104, 2939, 2869, 1726, 1707, 1494, 1454, 1376, 1269, 1221, 1107. Anal. Calc. for C29H30O3Fe*1/8(CH_2_Cl_2_): C, 69.61; H, 6.44. Found: C, 70.90; H, 6.54.

### 3.2. XRD Data Collection and Refinement

The X-ray diffraction data were collected on a Bruker APEX II Ultra diffractometer using Mo Kα radiation. The crystal data, data collection, and structure refinement details are summarized in Table 6.

### 3.3. Hirshfeld Surface Analysis

Hirshfeld surfaces (HS) and 2D fingerprint plots were generated using CrystalExplorer21 [14]. The Hirshfeld surface of ferrocene–hormone complex **2** was mapped over *dnorm* (−0.2740 to 1.4579 a.u.), de (0.9631 to 2.5646 a.u.), shape index (−1.0000 to 1.0000 a.u.), and curvedness (−4.0000 to 0.4000 a.u.).

### 3.4. Density Functional Theory Study

Gas-phase optimization and relaxed potential energy surface (PES) scan studies were performed using Gaussian16W [31] software. The crystal structures of **2** and **4** were used as starting points for the gas-phase optimization. The optimization of each structure was performed using the density functional theory (DFT) and B3LYP [32] hybrid density functional methods. Since derivative **2** contained a metal in its structure, both optimizations were performed with the LANL2DZ [33,34] level of theory as a basis set. Harmonic vibrational frequencies were calculated for the optimized structures to confirm that the geometries obtained were minima. No imaginary frequencies were found. The resulting optimized structures were then utilized to perform the relaxed PES scan studies. The C2-C1-O1-C19 dihedral angle was studied in 72 steps (at 5 degrees from each step) using the same parameters selected for the gas-phase optimization.

**Table 6 molecules-28-06147-t006:** Crystal data and experimental details.

Crystal Data	
Chemical formula	C_29_H_30_FeO_3_
M_r_	482.38
Crystal system	Monoclinic
Space group	C2
a, b, c (Å)	10.5771 (6), 11.1471 (7), 20.02457 (12)
β (°)	102.174 (1)
V (Å^3^)	2308.0 (2) Å^3^
Z	4
F (000)	1016
D(calc) (mg/m^3^)	1.388
Radiation type	Mo *K*α
μ (mm^−1^)	0.68
Crystal Size (mm)	0.28 × 0.26 × 0.24
T	100 K
**Data Collection**	
Diffractometer	Bruker *APEX* II Ultra
Absorption Correction	Multi-scan using SADABS2016/2 (Bruker, 2016/2)
*T*_min_, *T*_max_	0.695, 0.746
No. of measured, independent, and observed [*I* > 2σ(*I*)] reflections	19544, 5778, 5568
*R* _int_	0.030
**Refinement**	
*R*[*F*^2^ > 2σ(*F*^2^)], *wR*(*F*^2^), *S*	0.025, 0.062, 1.03
No. of reflections	5778
No. of parameters	299
No. of restraints	1
H-atom treatment	H-atom parameters constrained
Δρ_max_, Δρ_min_ (e Å^−3^)	0.33, −0.17
Absolute structure	Flack x determined using 2533 quotients [(I+) − (I−)]/[(I+) + (I−)] [35]
Absolute structure parameter	0.015 (5)

Computer programs used: APEX3 v2017.3-0 (Bruker, Bellirica, MA, USA, 2017), SAINT v8.38A (Bruker, 2017), SHELXS [36], XL [36], Olex2 [37].

### 3.5. Fluorescence Measurements

A stock solution of human serum albumin (10.0 µM) was prepared using phosphate-buffered saline (PBS; Sigma Aldrich, pH 7.4, liquid, sterile-filtered, suitable for cell culture), and a 500 µL aliquot of this stock solution was diluted in 4.5 mL of PBS. A volume of 2.5 mL of this 1 µM HSA solution was transferred to a 1.0 cm quartz cell and titrated using a 1 mM (1.5 mM for 30 °C) solution of 3-ferrocenyl-estra-1,3,5 (10)-triene-17-one (**2**) on DMSO. Each titration was prepared with 1 µL of the 2 solution, gently agitated, and incubated at the desired temperature for 20 min. Ten titrations were prepared for each temperature (20 °C, 25 °C, and 30 °C) to achieve a molar ratio of protein to ferrocene for complex **2** of 4:1 (1:5 for 30 °C). The quenching progress was monitored using a Jasco FP-8500 Spectrofluorometer. The fluorescence measurements were made using the parameters shown in Table 7.

### 3.6. Docking Studies

Docking Studies were performed using AutoDockVina [38] software. The crystal structure of HSA complexed with myristic acid (PDB ID:1BJ5) was used in this computational study [39]. Mystiric acid was eliminated, and missing fragments of some amino acid chains were added using Modeller 9.21 [40,41,42,43]. Polar hydrogens were then added to the protein structure, and Gasteiger charges were computed using AutoDockTools 1.5.6 [44]. The crystal structure of ferrocene conjugate **2** was obtained through X-ray diffraction studies.

Docking studies were performed at both drug-binding sites of HSA (sites I and II). Grid boxes were positioned at the center of each ligand-binding domain. At site I, the x-, y-, and z-centers were located at 35, 13, and 9.117, respectively. The spacing of the grid box was set to 1.00 Å, and the dimensions were set to 14 (size x), 20 (size y), and 18 (size z). Residues LEU219, PHE223, LEU238, HIS242, CYS245, ARG257, LEU260, ILE264, SER287, and ILE290 were selected as flexible. At site II, the x-, y-, and z-centers were located at 9.48, 2.812, and 19.939, respectively. The grid box dimensions were 29 (size x), 33 (size y), and 30 (size z). Residues SER489, PHE488, LEU460, LEU457, VAL415, LEU387, and TYR411 were selected as flexible.

## 4. Conclusions

The ferrocene–hormone complex 3-ferrocenyl-estra-1,3,5(10)-triene-17-one (**2**) was successfully synthesized and characterized using Nuclear Magnetic Resonance Spectroscopy, Fourier Transform Infrared Spectroscopy, and X-ray Diffraction (XRD). The crystal structure of **2** showed C-H···O hydrogen bonds and C-H···π bonds, which were closely inspected through Hirshfeld surface analysis. Density functional theory studies were performed to investigate the rotation of the C3-O1 single bond, and the rotational barrier was determined. The interaction of ferrocene–hormone complex **2** with human serum albumin was studied through fluorescence spectroscopy, which suggested the formation of the **2**–HSA complex through van der Waals and hydrogen bonding interactions. The in silico docking results were consistent with these results. The binding pose obtained showed possible hydrogen bonds with SER489 and C-H···π interactions with residues TYR411 and PHE488.

## Figures and Tables

**Figure 1 molecules-28-06147-f001:**
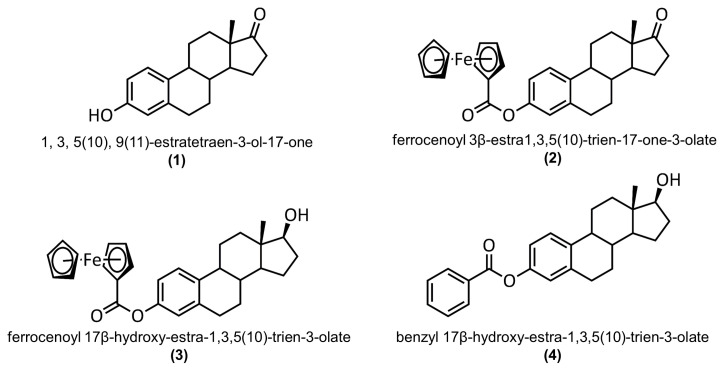
Structures of 3-hydroxy-1,3,5(10)-estratrien-17-one (**1**), 3-ferrocenyl-estra-1,3,5 (10)-triene-17-one (**2**), 3-ferrocenyl-estra-1,3,5 (10)-triene-17β-ol (**3**), and 3-benzyl-estra-1,3,5 (10)-triene-17β-ol (**4**).

**Figure 2 molecules-28-06147-f002:**
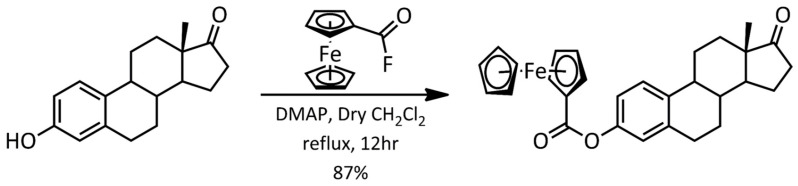
Reaction scheme for the synthesis of 3-ferrocenyl-estra-1,3,5(10)-triene-17-one (**2**).

**Figure 3 molecules-28-06147-f003:**
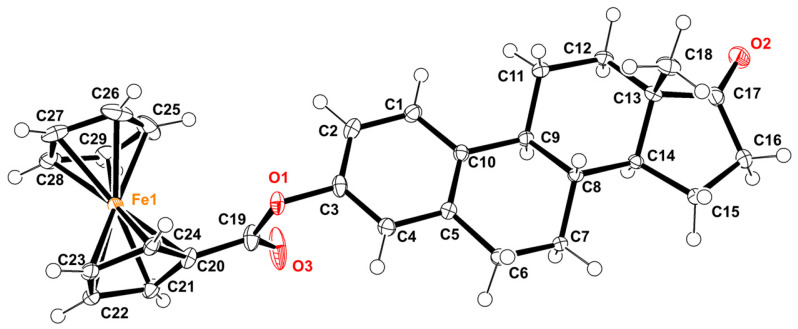
ORTEP diagram of 3-ferrocenyl-estra-1,3,5 (10)-triene-17-one, with displacement ellipsoids drawn at the 50% probability level.

**Figure 4 molecules-28-06147-f004:**
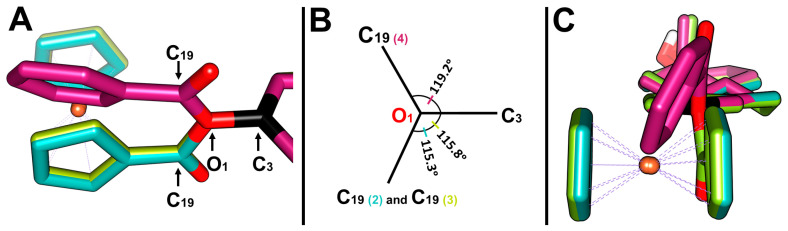
(**A**) Superposition on the C3-O1 bond of 3-ferrocenyl-estra-1,3,5 (10)-triene-17-one (**2**, cyan), 3-ferrocenyl-estra-1,3,5 (10)-triene-17β-ol (**3**, green), and 3-benzyl-estra-1,3,5 (10)-triene-17β-ol (**4**, magenta). (**B**) Angle ∠C3-O1-C19 of each complex relative to the C3-O1 bond. (**C**) Side representation of the dihedral angle between the cyclopentadienyl (**2** and **3**) or benzyl (**4**) planes and the aromatic group of the hormone moiety.

**Figure 5 molecules-28-06147-f005:**
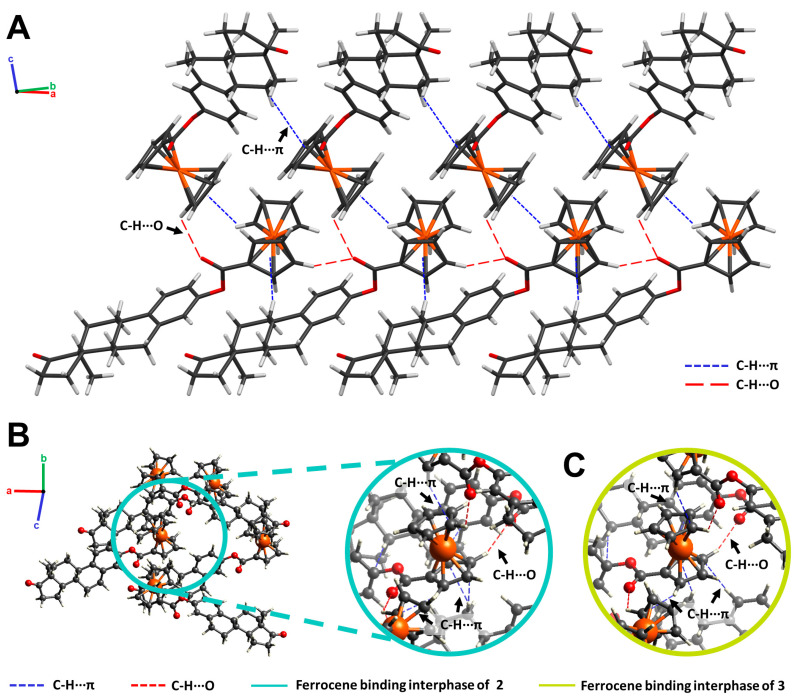
(**A**) Crystal packing of 3-ferrocenyl-estra-1,3,5 (10)-triene-17-one (**2**) projected along the (110) plane. C-H···π (blue) and C-H···O (red) interactions are illustrated as dashed lines. (**B**) Ferrocene-binding interphase of **2** (cyan). (**C**) Ferrocene-binding interphase of 3-ferrocenyl-estra-1,3,5 (10)-triene-17β-ol (**3**, green).

**Figure 6 molecules-28-06147-f006:**
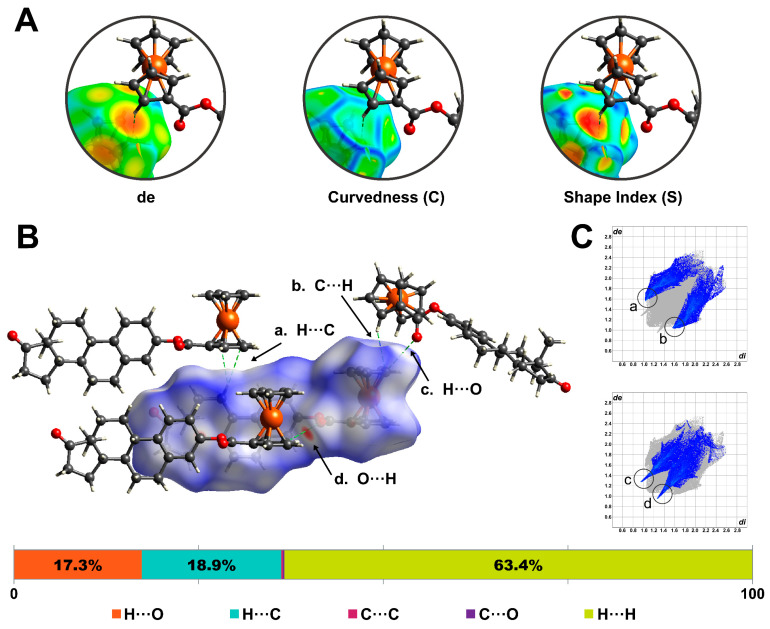
Hirshfeld surface was mapped over (**A**) distance external to the surface (*de*), shape-index curvedness, and (**B**) *dnorm*. (**C**) Two-dimensional fingerprint plots highlight the frequency and contributions of C-H···C and C-H···O contacts within the surface.

**Figure 7 molecules-28-06147-f007:**
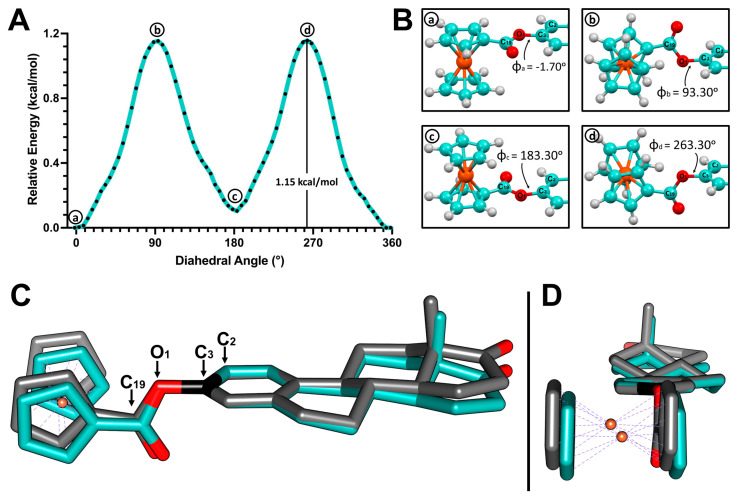
(**A**) Potential energy surface (PES) of the C3-O1 single bond in 3-ferrocenyl-estra-1,3,5 (10)-triene-17-one (**2**). (**B**) Structures corresponding to the local minima (**a**,**c**) and local maxima (**b**,**d**). (**C**) Superposition of the crystal structure of **2** (cyan) and the **b** local maximum structure (gray) obtained from the PES scan. (**D**) Side view.

**Figure 8 molecules-28-06147-f008:**
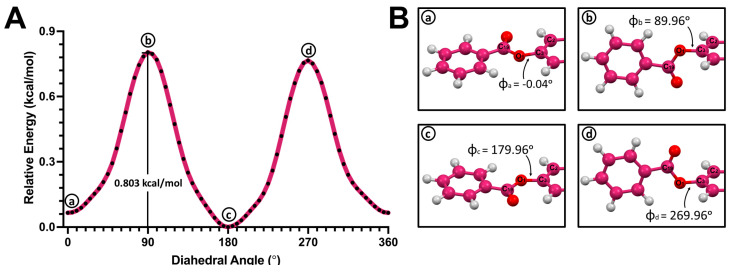
(**A**) Potential energy surface (PES) of the C3-O1 single bond in 3-benzyl-estra-1,3,5 (10)-triene-17β-ol (**4**). (**B**) Structures corresponding to the local minima (**a**,**c**) and local maxima (**b**,**d**).

**Figure 9 molecules-28-06147-f009:**
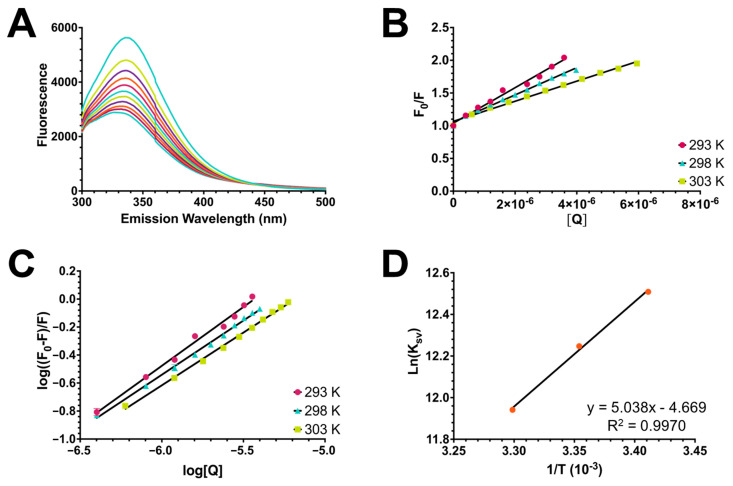
(**A**) Fluorescence quenching spectra of human serum albumin (HSA) in the presence of 3-ferrocenyl-estra-1,3,5 (10)-triene-17-one (**2**) studied at 303 K (**B**) Stern–Volmer plots for the fluorescence quenching of **2** and HSA at 293 K, 298 K, and 303 K. (**C**) The plots of *log* ((*F*_0_ − *F*)/*F*) versus *log* [*Q*] for the interaction between **2** and HSA at 293 K, 298 K, and 303 K. (**D**) Van’t Hoff plot for the binding of HSA to ferrocene–hormone complex **2**.

**Figure 10 molecules-28-06147-f010:**
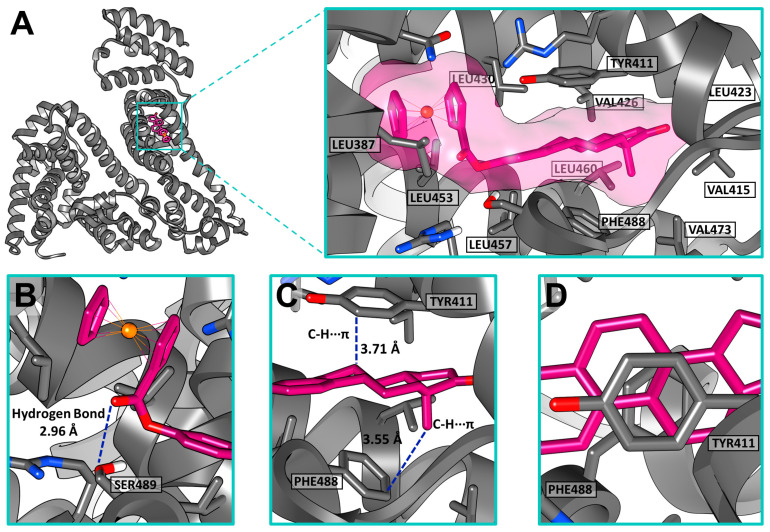
(**A**) Docking pose of 3-ferrocenyl-estra-1,3,5 (10)-triene-17-one (**2**) inside human serum albumin (HSA) drug-binding site II. (**B**) Hydrogen bonding interaction between SER489 and **2.** (**C**) C-H···π interactions with TYR411 and PHE488. (**D**) Top view.

**Table 1 molecules-28-06147-t001:** Selected geometrical parameters (Å, °) for compounds **1**, **2**, **3**, and **4**.

	1	2	3	4
**Bond lengths (Å)**				
Fe-C(Cp)_avg_		2.040 (13)	2.043 (13)	
Fe-C(Cp)*_subt_		2.018 (2)	2.018 (6)	
C(Cp)_subt_-C19		1.466 (3)	1.470 (1)	1.487 (4)
C19-O3		1.194 (3)	1.190 (1)	1.195 (3)
C19-O1		1.364 (3)	1.353 (9)	1.337 (3)
C3-O1	1.374 (2)	1.408 (2)	1.407 (9)	1.415 (3)
C17-O2	1.219 (2)	1.202 (3)		
**Angles (**°**)**				
C17-C13-C14	100.1 (1)	100.7 (2)	100.1 (5)	99.5 (2)
C3-O1-C19		115.3 (2)	115.8 (6)	119.2 (2)
O1-C19-C20		112.1 (2)	111.8 (7)	111.2 (2)
**Dihedral Angles (**°**)**				
Φ_1_ ^(i)^		97.5 (2)	96.8 (8)	−94.8 (3)
Φ_2_ ^(ii)^		175.5 (2)	176.7 (6)	166.3 (2)

Dihedral angles are defined as ^(i)^ C2-C3-O1-C19 and ^(ii)^ C3-O1-C19-C20.

**Table 2 molecules-28-06147-t002:** Hydrogen-bond geometry (Å, °). Cg_1,_ Cg_2,_ and Cg_3_ are the centroids of the C1–C6, C20–C24, and C25–C29 rings, respectively.

D-H···A	D-A (Å)	H···A (Å)	D···A (Å)	D-H···A (°)
C4-H4···O2 ^(i)^	0.95	5.613	3.469	150.1
C6-H6B···O2 ^(i)^	0.99	2.643	3.325	126.2
C28-H28···O3 ^(ii)^	0.95	2.614	3.210	149.6
C23-H23···O3 ^(iii)^	0.95	2.357	3.212	121.1
C24-H24···Cg_1_ ^(iii)^	0.95	3.269	3.942	129.46
C14-H14···Cg_1_ ^(iii)^	1.00	3.115	4.022	151.47
C11-H11B···Cg_2_ ^(i)^	0.99	2.816	3.745	156.55
C21-H21···Cg_3_ ^(ii)^	0.95	2.876	3.813	169.37

Symmetry codes: ^(i)^ −1 + x,y,z ^(ii)^ −1/2 + x, −1/2 + y, 1 − z ^(iii)^ −1/2 + x, 1/2 + y,z.

**Table 3 molecules-28-06147-t003:** Stern–Volmer quenching constants for the interaction between **2** and HSA at 293 K, 298 K, and 303 K.

*T* (K)	*K_SV_* (10^5^, L/mol)	*k_q_* (10^13^, L/mol s)	*R* ^2^
293	2.70652	2.70652	0.9864
298	2.08388	2.08388	0.9922
303	1.53492	1.53492	0.9913

**Table 4 molecules-28-06147-t004:** Binding constant (K) and number of binding sites (n) for the interaction between **2** and HSA at 293 K, 298 K and 303 K.

*T* (K)	*K* (10^−5^, L/mol)	*n*	*R* ^2^
293	2.71019	0.8407	0.9892
298	7.79830	0.7748	0.9957
303	12.41652	0.7537	0.9962

**Table 5 molecules-28-06147-t005:** Thermodynamic parameters of **2**–HSA interaction.

*T* (K)	Δ*H* (kJ/mol)	Δ*S* (J/mol)	Δ*G* (kJ/mol)	*R* ^2^
293	−41.8883	−38.8202	−30.5139	0.9970
298			−30.3198	
303			−30.1257	

**Table 7 molecules-28-06147-t007:** Fluorescence measurement parameters.

Light Source	Xe Lamp
Excitation wavelength	280.0 nm
Emission measurement range	300–600 nm
Excitation bandwidth	10 nm
Emission bandwidth	5 nm
Response	50 ms
Data interval	0.5 nm
Scan speed	2000 nm/min

## Data Availability

Not available.

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
