# Peer review of "3-Ferrocenyl-estra-1,3,5 (10)-triene-17-one: Synthesis, Crystal Structure, Hirshfeld Surface Analysis, DFT Studies, and Its Binding to Human Serum Albumin Studied through Fluorescence Quenching and In Silico Docking Studies"

_molecules, 2023, doi:10.3390/molecules28166147_

Round 1

Reviewer 1 Report

Line 1, it should be stated what type of manuscript it is, I suggest “Article”

Lines 4, 18, etc. why is  human serum albumin with capital letters?

Line 11, it should be Fe(C5H5)(C24H25O3)

Lines 64-65, those results (NMR, FT-IR) should be presented in this article, the spectra should be shown and the peaks should be assigned

Lines 315-318, how can I obtain the cif file of this structure? The deposition number to CCDC should be added, the cif and checkcif should be uploaded as SI to this study.

Line 337, it is “basis” set, not “basic” set

Line 337, why the Authors have used such a small basis set. I would recommend 6-311++G(d,p). This is now computationally affordable on most computers, especially for such small systems as in this study.

DFT calculations: were the harmonic frequencies calculated as well? This is crucial to confirm that the structures are in minima.

Line 358, the choice of HSA as a macromolecule for docking must be justified. Do you have any experimental indications that the studied compounds should bind with those particular proteins?

Line 361, it should be “charges”

How was the grid generated for docking (dimensions)?

What was the docking protocol?

The docking is very basic, taking into consideration that the authors have docked solely one compound. The docking studies should be strengthened by the MM/GBSA and molecular docking calculations to assess the dynamic stability of the complex. After all, this is just one compound.

Figure 7A, those values are not relative energies but direct values. Instead, the Authors should present the real relative energies, i.e. with the lowest one set to zero.

Lines 179-180, 192-193, if the barriers are so small, why the dynamic of this system is not observed?

Lines 218-223, the text is not correctly formatted here.

Table 5, were those results compared with ones from docking (delta G)? Why not?

Line 282, I still strongly suggest that docking to Site I should be performed as well, and the results should be compared, especially the deltaG.

Lines 286-287, a Figure should be created showing those protein-ligand interactions with all the aa listed included in the scheme

Conclusion section is missing in this study.

Author Response

Reviewer #1

  • Line 1, it should be stated what type of manuscript it is, I suggest “Article”

Reply: Corrected, thanks!

  • Lines 4, 18, etc. why is human serum albumin with capital letters?

Reply: Corrected, thanks!

  • Line 11, it should be Fe(C5H5)(C24H25O3)

Reply: Corrected, thanks!

  • Lines 64-65, those results (NMR, FT-IR) should be presented in this article, the spectra should be shown and the peaks should be assigned

Reply: 1H NMR, 13C NMR, and FT-IR were added to the supplementary material. Peaks were assigned.

  • Lines 315-318, how can I obtain the cif file of this structure? The deposition number to CCDC should be added, the cif and checkcif should be uploaded as SI to this study.

Reply: The cif file was deposited to CCDC and the deposition number was added to line 40 (CCDC: 228769). The checkcif file was added to the supplementary material on pages 5-8.

  • Line 337, it is “basis” set, not “basic” set

Reply: Corrected, thanks!

  • Line 337, why the Authors have used such a small basis set. I would recommend 6-311++G(d,p). This is now computationally affordable on most computers, especially for such small systems as in this study.

Reply: We repeated the DFT studies using the LANL2DZ level of theory for the optimization of derivative 3. We chose this basis set to make the results comparable with the ferrocene hormone complex 2, which has metal in its structure.

  • DFT calculations: were the harmonic frequencies calculated as well? This is crucial to confirm that the structures are in minima.

Reply: The harmonic frequencies were calculated and no imaginary frequencies were found on either of the structures.

  • Line 358, the choice of HSA as a macromolecule for docking must be justified. Do you have any experimental indications that the studied compounds should bind with those particular proteins?

Reply: Lines 200-207 were added to explain the importance of studying the HSA-2 complex by Fluorescence quenching studies and the possible binding pose by docking studies.

  • Line 361, it should be “charges”

Reply: Corrected, thanks!

  • How was the grid generated for docking (dimensions)?

Reply: The grid dimensions for each docking study were added on lines 362-363 and lines 365-366.

  • What was the docking protocol?

Reply: Lines 359-367 were added describing the details for the docking studies.

  • The docking is very basic, taking into consideration that the authors have docked solely one compound. The docking studies should be strengthened by the MM/GBSA and molecular docking calculations to assess the dynamic stability of the complex. After all, this is just one compound.

Reply: The docking studies presented in this study were intended to provide insights into the most favorable binding pose of the protein-ligand complex between HSA and the ferrocene-estrone conjugate. Although the ΔG value provided by AutoDock is often not comparable to real (empirical) values, the presented docking poses are well aligned, in terms of intermolecular contact, from the thermodynamic values obtained from the Van't Hoff plot approaches. Crystallographic data will confirm unequivocally the real binding position of the ligand within the HSA drug-binding sites.

  • Figure 7A, those values are not relative energies but direct values. Instead, the Authors should present the real relative energies, i.e. with the lowest one set to zero.

Reply: Corrected, thanks!

  • Lines 179-180, 192-193, if the barriers are so small, why the dynamic of this system is not observed?

Reply: The same conformation is shown for both ferrocene-hormone complexes. Showing that this conformation provides stability and maximizes the interactions within the crystal packaging in these crystallization conditions.

  • Lines 218-223, the text is not correctly formatted here.

Reply: Corrected, thanks!

  • Table 5, were those results compared with ones from docking (delta G)? Why not?

Reply: We decided not to compare ΔG from the fluorescence quenching with that obtained from the scoring function of the docking studies. Docking studies often correctly predict the bound conformations but are less accurate when calculating the ΔG of the system when compared with benchmark results.  (Int. J. Mol. Sci. 201617(2), 144;  J . Chem. Inf. Model. 2021, 61, 6, 2957–2966 ; Molecules 201419(7), 10150-10176.)

  • Line 282, I still strongly suggest that docking to Site I should be performed as well, and the results should be compared, especially the deltaG.

Reply: Docking studies on HAS Site I were performed with an affinity of -12.4. Figure S4 shows the binding pose.

  • Lines 286-287, a Figure should be created showing those protein-ligand interactions with all the aa listed included in the scheme

Reply: Figure 10 has been modified to illustrate these interactions. Through closer inspection of the binding pose, we found that the ferrocene hormone complex could form C-H···π interactions with residues TYR411 and PHE488.

  • Conclusion section is missing in this study.

Reply: Conclusion was added on lines 401-411.

Reviewer 2 Report

1. Irregular presentation of picture serial numbers.

2. The Fig. 5 lacks a legend representation and coordinated axis.

3. It is necessary to describe in more detail about the prospects for the biomedical application. 4. In addition, the format and some other minor errors in the article should be check carefully.

5. Please provide the IR and TGA data and discuss it.

6. updated the Density functional theory study, some work may be considered, such as J. Phys. Chem. A, 2019, 123, 6751−6760; Monatsh. Chem, 2019, 150, 1355–1364 and Inorg. Chim, Acta 546(2023)121297

7. The stability of the clusters in the solution also must be checked. It is not sufficient to check their stability in the solid state.

8. There are many methods for studying the complex -HSA interaction such as: Circular Dichroisms, Gel electrophoresis, Viscosity measurements. I recommend that some other methods add to the methods of interaction studies.

revise

Author Response

Reviewer #2

  • Irregular presentation of picture serial numbers.

Reply: Corrected. The figures are presented for the first time in a sentence (example: “The reaction scheme is shown in Figure 2.”). The specific parts of each figure are referenced in the text in parentheses (example: “(Figure 9B)”).

  • The Fig. 5 lacks a legend representation and coordinated axis.

Reply: The legend representation and coordinated axes have been added. Thanks!

  • It is necessary to describe in more detail about the prospects for the biomedical application. 4. In addition, the format and some other minor errors in the article should be check carefully.

Reply: Corrected, thanks!

  • Please provide the IR and TGA data and discuss it.

Reply: The FT-IR spectra are provided in the supplementary material (Figure S3).

  • updated the Density functional theory study, some work may be considered, such as J. Phys. Chem. A, 2019, 123, 6751−6760; Monatsh. Chem, 2019, 150, 1355–1364 and Inorg. Chim, Acta 546(2023)121297

Reply: Thanks for the suggested articles. DFT studies for derivative 3 were repeated using LANL2DZ level of theory.

  • The stability of the clusters in the solution also must be checked. It is not sufficient to check their stability in the solid state.

Reply: We are not clear what exactly the reviewer wants. Could you explain a bit more?

  • There are many methods for studying the complex -HSA interaction such as: Circular Dichroisms, Gel electrophoresis, Viscosity measurements. I recommend that some other methods add to the methods of interaction studies.

Reply: Unfortunately, we do not have access to the equipment to perform either of these studies, but we will take it into consideration for future projects. Thank you for your suggestion.

Round 2

Reviewer 1 Report

The Authors have revised and improved their manuscript. This version can be accepted.

Reviewer 2 Report

accept